# Outcomes and Risk Factors of Critically Ill Patients with Hematological Malignancy. Prospective Single-Centre Observational Study

**DOI:** 10.3390/medicina57121317

**Published:** 2021-11-30

**Authors:** Šarūnas Judickas, Raimundas Stasiūnaitis, Andrius Žučenka, Tadas Žvirblis, Mindaugas Šerpytis, Jūratė Šipylaitė

**Affiliations:** 1Department of Anaesthesiology and Intensive Care, Institute of Clinical Medicine, Faculty of Medicine, Vilnius University, Santariskiu Str. 2, 08661 Vilnius, Lithuania; mindaugas.serpytis@santa.lt (M.Š.); jurate.sipylaite@santa.lt (J.Š.); 2Faculty of Medicine, Vilnius University, M. K. Čiurlionio Str. 21/27, 03101 Vilnius, Lithuania; raimundas.stasiunaitis@gmail.com; 3Clinic of Internal Diseases, Family Medicine and Oncology, Institute of Clinical Medicine, Faculty of Medicine, Vilnius University, Santariskiu Str. 2, 08661 Vilnius, Lithuania; andrius.zucenka@santa.lt; 4Department of Mechanics and Material Engineering, Faculty of Mechanics, Vilnius Gediminas Technical University Vilnius, J. Basanaviciaus Str. 28, 03224 Vilnius, Lithuania; tadas.zvirblis@vilniustech.lt

**Keywords:** blood cancer, oncohematological patients, intensive care unit, outcome, Baltic region

## Abstract

*Background and Objectives*: Oncohematological patients have a high risk of mortality when they need treatment in an intensive care unit (ICU). The aim of our study is to analyze the outcomes of oncohemathological patients admitted to the ICU and their risk factors. *Materials and Methods*: A prospective single-center observational study was performed with 114 patients from July 2017 to December 2019. Inclusion criteria were transfer to an ICU, hematological malignancy, age >18 years, a central line or arterial line inserted or planned to be inserted, and a signed informed consent form. Univariate and multivariable logistic regression models were used to evaluate the potential risk factors for ICU mortality. *Results*: ICU mortality was 44.74%. Invasive mechanical ventilation in ICU was used for 55.26% of the patients, and vasoactive drugs were used for 77.19% of patients. Factors independently associated with it were qSOFA score ≥2, increase of SOFA score over the first 48 h, mechanical ventilation on the first day in ICU, need for colistin therapy, lower arterial pH on arrival to ICU. Cut-off value of the noradrenaline dose associated with ICU mortality was 0.21 μg/kg/min with a ROC of 0.9686 (95% CI 0.93–1.00, *p* < 0.0001). *Conclusions*: Mortality of oncohematological patients in the ICU is high and it is associated with progression of organ dysfunction over the first 48 h in ICU, invasive mechanical ventilation and need for relatively low dose of noradrenaline. Despite our findings, we do not recommend making decisions regarding treatment limitations for patients who have reached cut-off dose of noradrenaline.

## 1. Introduction

The incidence of malignancies is increasing in Lithuania [1] and more than one thousand people are diagnosed with blood cancer every year [2]. The mortality rate of oncohematological patients who need treatment in the intensive care unit (ICU) is high and may reach up to 80% [3]. It is worse compared to that of non-oncological patients [4] and to patients with solid tumors, also associated with a worse quality of life after treatment [5]. However, data show that survival of patients with hematological malignancies has increased dramatically over the last few decades [6,7,8] including patients admitted to the ICU [9]. It is better in Western Europe than in Eastern Europe [10,11,12] which is still suffering from less effective care, fewer resources allocated to heath care [13], limited access to innovative treatment options, higher out-of-pocket spending on health [14,15], and differences in the setting of end of life which makes comparison of survival difficult. Apart from that, our region is moving forward and the center of hematology, oncology and transfusion medicine of Vilnius university hospital Santaros klinikos is the largest center for bone marrow transplant in the Baltics (and one of the largest in Europe) [16]. Unfortunately, there are no data regarding mortality of critically ill patients with blood cancer in Baltics. The aim of our study is to analyze the outcomes of oncohemathological patients who are admitted to ICU and their risk factors.

## 2. Materials and Methods

### 2.1. Study Design

This prospective observational study took place at Vilnius University hospital Santaros Klinikos from July of 2017 to December of 2019. The decision to admit a patient to the ICU was made by the intensive care doctor in charge. There was no treatment limitations or withdrawal. None of these patients had a “do not resuscitate” status. The study was approved by Vilnius Regional Biomedical Research Ethics Committee.

### 2.2. Participants

Inclusion criteria were as follows: transfer to the ICU; hematological malignancy; age >18 years on the day of admission to the ICU; central line or arterial line inserted or planned to be inserted within 3 h after transfer to the ICU; and a signed informed consent form. If patients were unable to consent or sign informed consent form due to a clinical condition, their next-of-kin were approached regarding participation in this study. Malignancies were stratified to either standard or high risk based on the clinical, genetical, laboratorial, and radiological stratification parameters for different hematological diseases. All relapsed/refractory malignancies were considered as high risk. Chemotherapy regimens were classified as intensive (high doses of chemotherapy with highly possible toxicities and myelosuppression) or non-intensive (lower doses of chemotherapy or less toxic anticancer agents). Conditioning regimens before bone marrow transplantation (BMT) were considered to be intensive therapies. For classification of chemotherapy regimens please refer to Appendix A.

### 2.3. Research Data

Blood and urine samples were taken during the first 3 h after the patient was transferred to the ICU and repeated according to predefined plan (Appendix A). Charlson’s comorbidity index was calculated on arrival in the ICU. The SOFA score was calculated each day for the first 5 days starting from admission to the ICU. If a patient had multiple ICU admissions, only data from the first admission were analyzed. Neutropenia was defined as an absolute neutrophil count below 0.5 × 10^9^/L. Sepsis and septic shock were defined according to the SEPSIS-3 definition [17]. Organ failure was diagnosed if the SOFA score for that organ was 2 or more. Acute kidney injury was also diagnosed separately according to the KDIGO scoring system. The decision to start renal replacement therapy was made by the ICU doctor in charge. All of the continuous renal replacement therapies were performed using the Prismaflex system (Baxter Co. Deerfield, IL, USA) with citrate anticoagulation and CVVHDF mode. The effluent dose was prescribed by the attending ICU doctor. We recorded the highest dose of vasopressors which was given for at least 1 h anytime in ICU. Data were collected into an electronic predefined database from the electronic records and observation charts. Follow-up information was obtained from the patient’s electronic records. Figure 1 shows the flow diagram of the study.

### 2.4. Statistical Analysis

Kolmogorov–Smirnov test was used to assess the normality of the distribution. A Student’s *t*-test was used to evaluate the differences between the two independent normally distributed variables, while the Mann–Whitney U test was used for non-normally distributed variables. Fisher’s exact test was used to evaluate the differences between two independent categorical data groups. Univariate and multivariable logistic regression models were used to evaluate the potential risk factors for ICU mortality. Multivariable logistic regression was performed with two steps: first, we grouped factors which were found to be statistically significant in univariate logistic regression into three groups: scoring systems, therapy and laboratory tests. After that, multivariable logistic regression was performed in each group and only factors which remained statistically significant have been selected and entered in the final (second step) multivariable logistic regression which is presented in results. Odds ratios and their 95% confidence intervals were also calculated. Youden’s J statistic with receiver operating characteristic (ROC) analysis was used for cut-off values. Survival was estimated using the Kaplan–Meier method. A two-tailed *p*-value less than 0.05 was considered to be statistically significant. Statistical analysis was performed using the Statistical Analysis System (SAS Institute, Cary, NC, USA) package version 9.2.

## 3. Results

The main patient characteristics are presented in Table 1. The most common reasons for ICU admission were acute respiratory failure (*n* = 48, 42.11%) and shock (*n* = 23, 20.18%). The main hematological malignancies were acute myeloid leukemia (*n* = 46, 40.35%) and non-Hodgkin’s lymphomas (*n* = 28, 24.6%). BMT was performed for 39 patients (34.21%). Neutropenia on admission to ICU was observed in 44 patients (38.60%). SOFA scores in ICU and their changes are reported in Appendix A. Invasive mechanical ventilation in ICU was used for more than half of the patients (*n* = 63, 55.26%), and for one-third of the patients, it was started on the first day in the ICU (*n* = 37, 32.46%). Vasoactive drugs were used for 88 patients (77.19%). Acute renal injury on admission to ICU was diagnosed in 28.95% patients (*n* = 33). Continuous renal replacement therapy was used for 29 patients (25.4%). Table 2 summarizes the characteristics of patients who survived or died in the ICU.

Non-survivors had higher SOFA scores upon admission to the ICU (Figure 2a) which increased more often over the first 48 h in ICU (Figure 2b) compared with survivors. ICU, 30-day, 90-day mortality rates were 44.74%, 54.39% and 64.91% respectively, as of 28 May 2020. Mortality predicted using the APACHE II and SAPS 3 scores was lower compared with the observed mortality (44.75% vs. 54.39%, *p* < 0.0001 and 44.68% vs. 54.39%, *p* < 0.0001, respectively). Factors that were statistically significantly associated with higher risk of ICU mortality in the univariate analysis are presented in Table 3. After multivariable adjustment, the independent factors that were associated with ICU mortality were qSOFA ≥ 2, increasing SOFA score over the first 48 h in the ICU, invasive mechanical ventilation on day 1 in the ICU, need for colistin therapy in the ICU, and lower arterial pH upon arrival in the ICU (Table 4). We further analyzed patients receiving vasoactive medications. To avoid the cumulative effect of vasoactive medications, we analyzed patients who only received noradrenaline (*n* = 59, 51.75%) and calculated the area under the ROC curve (AUROC) to analyze the discriminative ability of noradrenaline to predict ICU mortality and to select a cut-off value for noradrenaline. The optimal cut-off value (Youden Index = 91) for the noradrenaline dose was 0.212 μg/kg/min. The AUROC was 0.9686 (95% CI 0.9291–1.0000, *p* < 0.0001), sensitivity was 94.1%, and specificity was 97.1% (Figure 3). All of the patients with the maximum noradrenaline dose of >0.212 μg/kg/min died in the ICU.

## 4. Discussion

Our observed ICU, 30-day and 90-day mortality was 44.74%, 54.39%, and 64.91%, respectively. These results are similar to previous studies, and mortality is lower than it was in multicenter retrospective study of 200 patients with hematological malignancies made in Poland (ICU mortality, 67%; hospital mortality, 78.8%) [10] and single center prospective observational study of 170 patients with hematological malignancies treated in ICU in Croatia (ICU mortality, 53.5%) [11], but the reported data are limited to only a few centers. Better results were found in a prospective study of 1011 critically ill patients with hematologic malignancies from 17 centers in France and Belgium; hospital mortality was 39.3%, and 90-day mortality was 47.5% and 56.7%, respectively [18]. In order to compare survival rates between countries correctly, we should be able to match patients, acceptance criteria, treatment options, and their choices including treatment limitation or withdrawal of treatment. We did not set any limits for the maximum noradrenaline dose which is seen in another centers. Our study also included patients with poor performance status and uncontrolled or refractory graft versus host disease (GVHD), which might have compromised the acceptance of these patients to the ICU in another countries.

Patients who have undergone a BMT have a high mortality rate, which is associated with the malignancy and with severe life-threatening post-transplant complications. Neither BMT nor its type (i.e., allogenic or autologous BMT) or intensity of chemotherapy affected ICU mortality in our study. Our study included 16 (14.04%) patients with GVHD, which was uncontrolled or refractory in nine patients and all of them died within 30 days after admission to the ICU. Our findings are consistent with the recent study [19], where the 90-day survival of patients with uncontrolled acute GVHD was only 25.8%.

The qSOFA score is a simple tool that was made to suspected sepsis in non-ICU patients. However, the latest surviving sepsis campaign guidelines recommend against using qSOFA as a single screening tool for sepsis or septic shock [20]. Despite this, we found that a qSOFA score ≥ 2 was independently associated with ICU mortality. In a study that analyzed AML patients with febrile neutropenia, a qSOFA score ≥ 2 was also associated with infectious mortality and was a good predictor of mortality when combined with C-reactive protein [21]. A study of cancer patients that included 23.4% of patients with hematological malignancies found that a qSOFA score ≥ 2 is associated with higher risk of hospital mortality and a prolonged ICU stay [22]. However, the SOFA score outperformed the qSOFA score in predicting mortality for immunocompromised patients [23,24].

In a multivariable analysis, an increase in the SOFA score during the first 48 h in the ICU was independently associated with ICU mortality. Our findings are consistent with the literature. The SOFA score upon ICU admission was associated with mortality [18] and all patients with a SOFA score of 15 or higher died in the ICU [25]. Additionally, a higher SOFA score on admission and worsening organ dysfunction over the first three days were both independently associated with mortality [26] together with no change in the SOFA score [27].

The use of vasoactive drugs is accepted as an independent risk factor for mortality in oncohematological patients [11,28]. We found that it was associated with ICU mortality in a univariate analysis and cut-off value for the maximum noradrenaline dose was 0.212 μg/kg/min. This dose is low compared to the usual doses that intensivists administer in their daily practice and most importantly it shows that critically ill oncohaematological patients are extremely fragile—fatality of patients who reached this dose was 100%. To the best of our knowledge, this is the first prospective study that analyzed the cut-off value of the noradrenaline dose in critically ill oncohematological patients. There are studies which analyzed the effect of higher noradrenaline dose [29]. For patients with septic shock and noradrenaline dose ≥ 0.3 μg/kg/min 28-day mortality varied from 48.5% to 72.2%, and only 17% of patients had hematological comorbidity [30]. In another study patients received a vasopressor dose of at least 1 μg/kg/min for more than 1 h, and a cut-off value associated with mortality which was 0.75 μg/kg/min. Additionally, 35% of patients were immunocompromised, but the authors do not provide data regarding hematological malignancies [31]. It was found that all critically ill patients with febrile neutropenia who received a noradrenaline dose that was higher than 0.1 μg/kg/min died in the ICU [3]. The data from immunocompromised patients with septic shock shows that a high mortality is associated with much lower doses of vasopressors. High exogenous noradrenaline doses may also have deleterious consequences such as myocardial cell injury, oxidative stress, alteration of sepsis-associated immunomodulation and immunoparalysis [32]. In vitro and animal studies found administration of high noradrenaline doses to be anti-inflammatory and to directly promote bacterial growth. Neutrophils incubated with noradrenaline displayed an immunosuppressive phenotype [33]. Despite our findings, we do not recommend making decisions regarding treatment limitations for patients who have reached cut-off dose of noradrenaline. When in septic shock vasopressor doses are increasing or if the dose remains high, we should always rule out other reasons that might compromise hemodynamics such as myocardial damage, volume depletion, vascular compression by a mass, thrombus, tamponade, and pneumothorax.

The use of invasive mechanical ventilation in oncohematological patients is an independent predictor of mortality [11,18,28,34,35]. In the univariate and multivariable analyses, even invasive mechanical ventilation on the first day in the ICU was independently associated with mortality. In our study, invasive mechanical ventilation was used for 63 patients (55.26%) during their stay in the ICU, and mortality was 77.78%. These findings are comparable to other studies in which mortality ranged from 35% to 70% [36], with an average of 60.5% [18,37].

The use of colistin therapy was independently associated with ICU mortality in our study. We suggest that it should be interpreted as a surrogate of infection caused by *A. baumannii* which is a frequent pathogen in hospital-acquired infections in our unit. During this study, this microorganism was sensitive to colistin, and we used this therapy for patients with highly suspected or confirmed infection that was caused by *A. baumannii*.

The limitation of our study is that it is a single-center experience. Our center is the largest in the region, but we did not include patients from other hospitals with hematological facilities in Lithuania (one of which is also performing bone marrow transplantation). Despite the hematological diagnosis, patients were different in terms of genetics, bone marrow recovery, colonization of multi-drug resistant microorganisms, and infective complications, which makes their comparison difficult.

## 5. Conclusions

The ICU mortality of oncohematological patients in our study was 44.74%. Factors that were independently associated with it were a qSOFA score ≥ 2, increase in the SOFA score over the first 48 h in ICU, need for invasive mechanical ventilation on the first day in the ICU, need for colistin therapy, and lower arterial pH upon arrival in the ICU. We found that a low cut-off value for the noradrenaline dose was associated with ICU mortality. Thus, we suggest that condition of oncohaematological patients is more critical than it appears and that these patients need to be transferred to ICU before it becomes too critical.

## Figures and Tables

**Figure 1 medicina-57-01317-f001:**
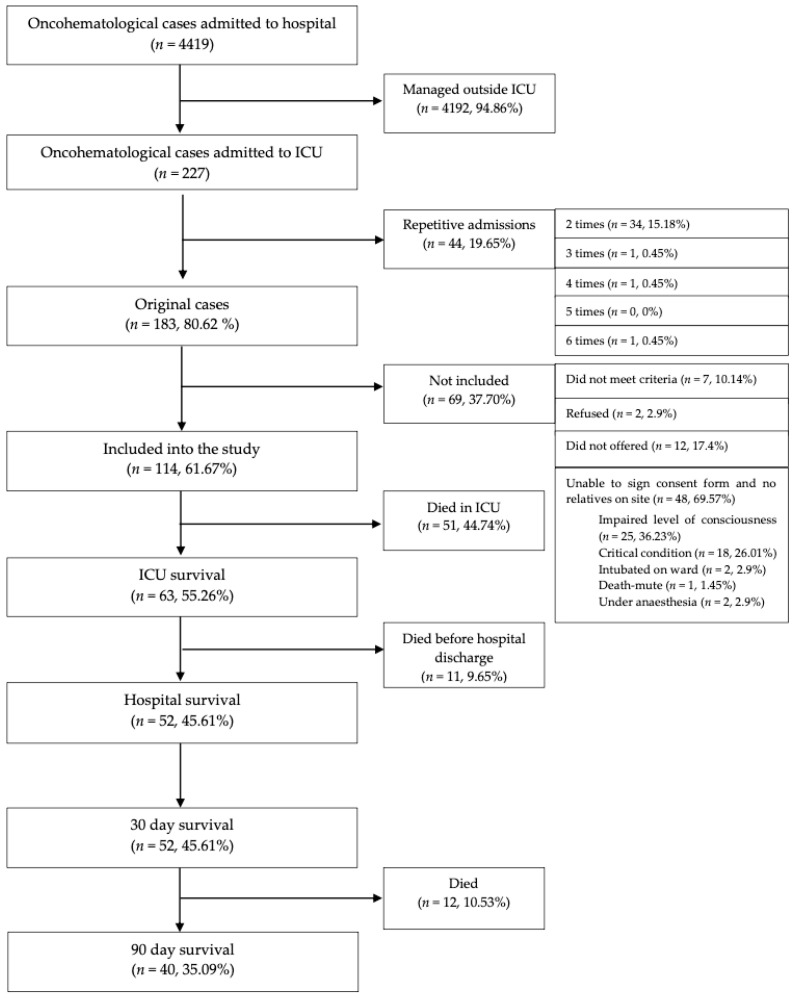
Study flow diagram.

**Figure 2 medicina-57-01317-f002:**
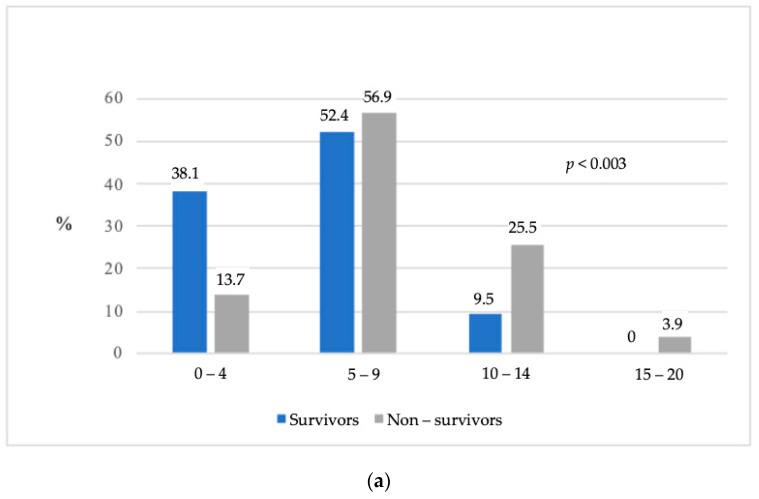
SOFA score in the ICU. (**a**) SOFA score on admission to the ICU; (**b**) dynamics of the SOFA score during the first 48 h in the ICU.

**Figure 3 medicina-57-01317-f003:**
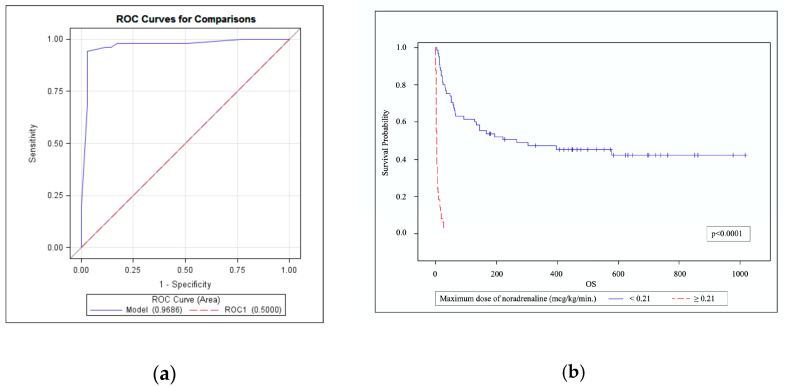
Noradrenaline dose and mortality. (**a**) ROC curve for the cut-off value of the noradrenaline dose associated with ICU mortality. AUROC 0.9686 (95% CI 0.9291–1.0000, *p* < 0.0001), sensitivity 94.1%, specificity 97.1%; (**b**) Kaplan–Meier curves for overall survival of patients with a noradrenaline dose < 0.21 μg/kg/min or ≥0.21 μg/kg/min.

**Table 1 medicina-57-01317-t001:** Patient characteristics.

Age, years (mean ± SD).	59.8 ± 15.38
Male sex (n (%))	49 (57.0)
Source of admission (n (%))	
Ward	92 (80.7)
Operating theatre	6 (5.26)
Emergency department	16 (14.04)
qSOFA score (mean ± SD)	1.4 ± 0.91
APACHE II score (mean ± SD)	21.72 ± 5.68
SAPS 3 score (mean ± SD)	75.01 ± 13.27
SOFA score on admission to ICU (mean ± SD)	6.56 ± 3.20
Charlson’s comorbidity index (mean ± SD)	4.90 ± 2.26
ECOG ≤ 2 (n (%))	85 (74.56)
Hematological diagnosis (n (%))	
Acute Myeloid Leukaemia	46 (40.35)
Non-Hodgkin’s Lymphoma	28 (24.6)
Multiple Myeloma	13 (11.40)
Chronic Lymphocytic Leukaemia	11 (9.65)
Acute Lymphoblastic Leukaemia	8 (7.0)
Hodgkin’s Lymphoma	4 (3.5)
Other	4 (3.5)
Graft versus host disease	16 (14.04)
Controlled or stable	7 (6.14)
Uncontrolled	7 (6.14)
Refractory	2 (6.14)
Chemotherapy intensive regimen (n (%))	56 (49.12)
Bone marrow transplantation (n (%))	39 (34.21)
Autologous	14 (12.28)
Allogenic	25 (21.93)
Reason for ICU admission (n (%))	
Acute respiratory failure	48 (42.11)
Shock	23 (20.18)
Neurological impairment	14 (12.28)
Sepsis	7 (6.14)
Multiple organ failure	6 (5.26)
Observation after surgery	5 (4.39)
Post cardiac arrest	2 (1.75)
Other	9 (7.89)
Length of stay before ICU admission, days (n (%))	20.65 ± 34.60
Management during ICU stay (n (%))	
Invasive mechanical ventilation 1st day	37 (32.46)
Invasive mechanical ventilation	63 (55.26)
Vasoactive drugs	88 (77.19)
CVVHDF	29 (25.4)
Length of stay in ICU, days (mean ± SD)	6.70 ± 5.48
Median follow-up (IQR), days	539.5 (367)

SD: standard deviation; n/a: not applicable; ICU: intensive care unit; qSOFA: quick sepsis related organ failure assessment; ECOG: Eastern Cooperative Oncology Group performance status; GVHD: graft versus host disease; CVVHDF: continuous veno-venous hemodiafiltration; IQR—interquartile range. SAPS 3 and APACHE II scores were calculated within the first 24 h after admission to the ICU. If the FiO_2_ was ≥50%, we calculated alveolar–arterial gradient using geographical atmospheric pressure for Vilnius, which was 750.025 mmHg (114 m altitude above sea level and temperature of 22 °C, which was room temperature in the ICU). QSOFA was calculated on admission to ICU.

**Table 2 medicina-57-01317-t002:** Patient characteristics according to ICU mortality.

Characteristics	Survivors (*n* = 63), *n* (%) 63 (55.26)	Non-Survivors (*n* = 51), *n* (%) 51 (44.74)	*p*-Value
Sex, female	30 (47.6)	19 (37.3)	0.342
Hematological diagnosis			0.474
Acute leukaemia	30 (47.6)	24 (47.1)	
Non-Hodgkin’s Lymphoma	14 (22.2)	14 (27.5)	
Multiple Myeloma	10 (15.9)	3 (5.9)	
Chronic leukaemia	5 (7.9)	6 (11.8)	
Hodgkin’s Lymphoma	1 (1.6)	3 (5.9)	
Other	3 (4.76)	1 (2.0)	
High-risk malignancy	43 (68.3)	36 (70.6)	0.840
Intensive chemotherapy	30 (47.6)	26 (51.0)	0.557
Bone marrow transplant	22 (34.9)	17 (33.3)	1.000
Allogenic	11 (17.5)	14 (27.5)	0.116
Autologous	11 (17.5)	3 (5.9)	
ECOG group			0.829
0–2	46 (73.0)	39 (76.5)	
≥3	17 (27.0)	12 (23.5)	
qSOFA score			0.004
0	12 (19.0)	4 (7.8)	
1	36 (57.1)	18 (35.3)	
2	10 (15.9)	17 (33.3)	
3	5 (7.9)	12 (23.5)	
Mechanical ventilation day 1 in ICU	9 (14.3)	28 (54.9)	<0.001
Mechanical ventilation anytime in ICU	14 (22.2)	49 (96.1)	<0.001
Vasoactive drugs anytime in ICU	35 (55.6)	51 (100.0)	<0.001
Renal replacement therapy	11 (17.5)	21 (41.2)	0.007
CVVHDF	9 (14.3)	20 (39.2)	0.004
Need for colistin therapy in ICU	6 (9.5)	14 (27.5)	0.024
Neutrophil count < 500/mm^3^	24 (38.1)	20 (39.2)	1.000
Source of admission to ICU			0.783
Emergency department	8 (12.7)	8 (15.7)	
Ward	44 (69.8)	33 (64.7)	
n.a.	11 (17.46)	10 (19.61)	

ECOG: Eastern Cooperative Oncology Group performance status; qSOFA: quick sepsis related organ failure assessment; ICU: intensive care unit; CVVHDF: continuous veno-venous hemodiafiltration; n.a.: not applicable.

**Table 3 medicina-57-01317-t003:** Univariate analysis of mortality in ICU.

Variable	Odds Ratio (95% CI)	*p* Value
Age, years	1.000 (0.976–1.025)	0.986
Female sex	1.531 (0.721–3.250)	0.267
Days in hospital before admission to ICU	0.994 (0.981–1.006)	0.327
ECOG	1.011 (0.730–1.399)	0.948
Charlson’s comorbidity index	1.050 (0.891–1.237)	0.563
High risk haematological malignancy	1.116 (0.500–2.491)	0.788
Intensive chemotherapy	1.277 (0.583–2.798)	0.541
Bone marrow transplantation	0.932 (0.428–2.031)	0.859
Autologous bone marrow transplantation	0.329 (0.085–1.275)	0.132
Allogenic bone marrow transplantation	1.535 (0.617–3.817)	
qSOFA ≥ 2	4.217 (1.891–9.405)	<0.001
SOFA score 5–9 on day 1 in ICU	3.013 (1.132–8.017)	0.004
SOFA score 10–20 on day 1 in ICU	8.571 (2.414–30.429)	
Equal SOFA score in the first 48 h in ICU	1.696 (0.384–7.489)	0.004
Increased SOFA score in the first 48 h in ICU	5.700 (1.748–18.587)	
APACHE II score	1.092 (1.019–1.171)	0.013
SAPS 3 score	1.041 (1.010–1.074)	0.010
Neutrophil count < 500/mm^3^ on arrival at the ICU	0.995 (0.464–2.130)	0.989
Haemoglobin on arrival at the ICU, g/L	1.026 (1.002–1.050)	0.030
Potassium on arrival at the ICU, mmol/L	2.244 (1.428–3.527)	<0.001
apH on arrival at the ICU, units	<0.001 (<0.001–0.031)	<0.001
Lactate on arrival at the ICU, mmol/L	1.314 (1.084–1.592)	0.005
Base excess on arrival at the ICU, units	0.918 (0.869–0.970)	0.002
Bicarbonate on arrival at the ICU, units	1.102 (1.020–1.190)	0.014
Need for colistin therapy in the ICU	3.531 (1.245–10.014)	0.018
Mechanical ventilation day 1 in the ICU	7.304 (2.983–17.888)	<0.001
Mechanical ventilation anytime in the ICU	85.749 (18.501–397.43)	<0.001
FiO_2_, %	1.032 (1.001–1.064)	0.042
Vasoactive drugs in ICU	2.213 (1.511–3.243)	<0.001
CVVHDF	3.870 (1.570–9.540)	0.003

ICU: intensive care unit; ECOG: Eastern Cooperative Oncology Group performance status; qSOFA: quick sepsis related organ failure assessment; SOFA: Sequential Organ Failure Assessment; APACHE: Acute Physiology and Chronic Health Evaluation; SAPS: The Simplified Acute Physiology Score; apH: arterial blood gas power of hydrogen; FiO_2_: fraction of inspired oxygen; CVVHDF: continuous veno-venous hemodiafiltration.

**Table 4 medicina-57-01317-t004:** Multivariable logistic regression analysis of mortality in the ICU.

Variable	Odds Ratio (95% CI)	*p* Value
qSOFA ≥ 2	4.403 (1.376–14.081)	0.0125
Equal SOFA score first 48 h in the ICU	4.903 (0.643–37.397)	0.0156
Increased SOFA score first 48 h in the ICU	11.171 (2.072–60.226)	
Invasive mechanical ventilation day 1 in the ICU	6.157 (1.867–20.308)	0.0028
Need for colistin therapy in the ICU	11.037 (2.673–45.572)	0.0009
Arterial pH on arrival to the ICU, units	0.392 (0.201–0.7620)	0.0058

QSOFA: quick sepsis related organ failure assessment; SOFA: sequential organ failure assessment; ICU: intensive care unit; apH: arterial blood gas power of hydrogen.

## Data Availability

The data that support the findings of this study are available on request from the corresponding author. The data are not publicly available due to the privacy restrictions stated in agreement for ethical approval.

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
