# Peer review of "Outcomes and Risk Factors of Critically Ill Patients with Hematological Malignancy. Prospective Single-Centre Observational Study"

_medicina, 2021, doi:10.3390/medicina57121317_

Round 1

Reviewer 1 Report

Thank you for giving me the opportunity to review this manuscript, which reports on a prospective observational study on the mortality among onco-hematological ICU patients in an academic medical center in Lithuania. Please find my comments below which I hope are beneficial.

Major issues:

  1. Introduction: The authors describe the knowledge gap well (lack of mortality data of onco-hematological ICU patients in the Baltics), but the authors should add a short reasoning as to why the mortality rate in the Baltics (or in general Eastern Europe) is expected to be different compared to settings we know more about (e.g., Western Europe or USA). In other words: Why can’t we just extrapolate from other settings? This would strengthen the motivation of the study.
  2. Methods: Subsections would ease reading of this section.
  3. Methods: The authors should provide more detail on what they considered high risk and standard risk malignancies.
  4. Methods: In the statistical analysis, the authors state that they first used univariate followed by multivariate logistic regressions. Later, they state they used Cox regression and report HRs. The terminology appears confusing to me: 1) The authors should check whether they performed a multivariate or multivariable regression (see 10.2105/AJPH.2012.300897). 2) Logistic regression yields ORs and does not comprehend the full time information of a time to event analysis, whereas Cox regression incorporates this time information to report HRs. I suggest that the authors disentangle the terminology in their analysis to describe what type of analysis they performed.
  5. Methods: The authors indicated that they used stepwise regression with backward selection. Why did the authors not simply do a multivariable regression (which might be sufficient)? In other words, what is the advantage of using the complex stepwise regression approach? By that end, when the authors stick to the stepwise regression, I suggest that the authors report which variables were omitted, what the goodness of fit of the model is, and how they addressed the issue of overfitting? On a side note, do two variables in Table 4 (equal SOFA and increased SOFA) have exactly the same p value?
  6. Methods/Results: The flowchart in Figure 1 and characteristics in Table 1 should be moved to the results section.
  7. Methods: Last-patient-in was reported to be Dec 2019, and mortality is reported as of 28 May 2020, which is less than 365 days after; however, the authors report a 365-day mortality. How many patients were followed for 365 days and how many patients less than that? I suggest to either report the number of patients followed <365 days or leave out the 365-days mortality (e.g., in favor of 6-months mortality if the authors have data on this from almost every patient).
  8. Discussion: At the beginning of the discussion, the authors compare their mortality rates with the rates from other studies. Did the other studies also consider only onco-hematological ICU patients? Were those multi-center, prospective studies? How many patients were analyzed? To put the numbers into context, the reader needs more information about the studies that the authors’ results are compared to.
  9. Disucssion: The authors state that other studies examined the effect of higher noradrenaline doses on mortality (references 23-25). Were these studies on onco-hematological patients only, or on general ICU patients?

Minor issues:

  1. Figure 1: In the upper two boxes, the authors state “patients admitted to hospital/ICU”, but as re-admissions are included, we suggest to re-name this as “cases admitted”.
  2. Figure 1: We suggest to re-arrange the mortality boxes and include them in the flowchart (the way they are arranged right now leaves them unrelated to the patient flow): There should be a vertical arrow from the box “original cases” down to a new box “survived at least 365 days” (or similar). The mortality boxes should then indicate how many patients leave the flowchart along this way (with horizontal arrows to each mortality box).
  3. Table 1: The authors report mean APACHE II, SAPS III and qSOFA scores, but do they represent the mean over all measurements (multiple measurements per patient)? This requires explanation in the Table or its caption.
  4. Results: I assume that the authors selected the optimal cut-off noradrenaline dose based on the highest Youden index. If so, the authors should state this in the results section (and also report the value of the Youden index).

Author Response

Response to Reviewer 1 Comments

Dear Sir or Madam,

We appreciate the time and effort that you have dedicated to providing your valuable feedback on our manuscript. We are grateful to you for your insightful comments on our paper. We have been able to incorporate changes to reflect the suggestions provided by you. The changes made within the manuscript are seen in “Track changes”. Here is a point-by-point comments and concerns.

Major issues:

Point 1. Introduction: The authors describe the knowledge gap well (lack of mortality data of onco-hematological ICU patients in the Baltics), but the authors should add a short reasoning as to why the mortality rate in the Baltics (or in general Eastern Europe) is expected to be different compared to settings we know more about (e.g., Western Europe or USA). In other words: Why can’t we just extrapolate from other settings? This would strengthen the motivation of the study.

Response 1: Thank you for pointing this out. We updated text with information about different setting of medicine in Eastern Europe compared with Western world together with 3 references. Now the text is as follows:

„The incidence of malignancies is increasing in Lithuania [1] and more than one thousand of people are diagnosed with blood cancer every year [2]. The mortality rate of oncohematological patients who need treatment in the intensive care unit (ICU) is high and it may reach up to 80% [3]. It is worse compared to that of non-oncological patients [4] and to patients with solid tumours, also associated with a worse quality of life after treatment [5]. However, data show that survival of patients with hematological malignancies has increased dramatically over the last decades [6–8] including patients admitted to the ICU [9]. It is better in Western Europe than in Eastern Europe [10–12] which is still suffering from less effective care, fewer resources allocated to heath care [13], limited access to innovative treatment options, higher out-of-pocket spending on health [14,15] and differences in the setting of end of life which makes comparison of survival difficult. Apart from that our region is moving forward and Centre of hematology, oncology and transfusion medicine of Vilnius university hospital Santaros klinikos is the largest centre for bone marrow transplant in Baltics and one of the largest in Europe [16]. Unfortunately, there is no data regarding mortality of critically ill patients with blood cancer in Baltics. The aim of our study is to analyse the outcomes of oncohemathological patients who are admitted to ICU and their risk factors.”

  1. M. Mounier, N. Bossard, L. Remontet, A. Belot, P. Minicozzi, R. De Angelis, R. Capocaccia, J. Iwaz, A. Monnereau, X. Troussard, M. Sant, M. Maynadié, and R. Giorgi, The Lancet Haematology 2, e481 (2015).
  2. P. Minicozzi, P. M. Walsh, M.-J. Sánchez, A. Trama, K. Innos, R. Marcos-Gragera, N. Dimitrova, L. Botta, T. B. Johannesen, S. Rossi, and M. Sant, European Journal of Cancer 93, 127 (2018).
  3. Organization for Economic Co-operation and Development Health spending (2021) (indicator). Available online: https://data.oecd.org/healthres/health-spending.htm (Accessed on 21 November 2021). DOI: 10.1787/8643de7e-en
  4. M. Tambor, J. Klich, and A. Domagała, IJERPH 18, 1382 (2021). Financing Healthcare in Central and Eastern European Countries: How Far Are We from Universal Health Coverage?

Point 2. Methods: Subsections would ease reading of this section.

Response 2: Thank you for great suggestion. We updated text with subsections.

Point 3: Methods: The authors should provide more detail on what they considered high risk and standard risk malignancies.

Response 3: Thank you very much for highlighting this issue. We could be more accurate in our manuscript regarding the expression of risk stratification for hematological malignancies. Malignancies were stratified to either standard or high risk based on the clinical, genetical, laboratorical and radiological stratification parameters for different hematological diseases (1-9). All relapsed/refractory malignancies were considered as high risk.

We updated manuscript with this text excluding references.

  1. Döhner H, Estey E, Grimwade D, Amadori S, Appelbaum FR, Büchner T, et al. Diagnosis and management of AML in adults: 2017 ELN recommendations from an international expert panel. Blood. 2017;129(4):424–47.
  2. Toft N, Birgens H, Abrahamsson J, Griškevičius L, Hallböök H, Heyman M, Klausen TW, Jónsson ÓG, Palk K, Pruunsild K, Quist-Paulsen P, Vaitkeviciene G, Vettenranta K, Åsberg A, Frandsen TL, Marquart HV, Madsen HO, Norén-Nyström U, Schmiegelow K. Results of NOPHO ALL2008 treatment for patients aged 1-45 years with acute lymphoblastic leukemia. Leukemia. 2018 Mar;32(3):606-615. doi: 10.1038/leu.2017.265. Epub 2017 Aug 18. PMID: 28819280.
  3. Palumbo A, Avet-Loiseau H, Oliva S, et al. Revised international staging system for multiple myeloma: a report from International Myeloma Working Group. J Clin Oncol. 2015;33:2863-2869.
  4. Jerusalem G, Beguin Y, Fassotte MF, Najjar F, Paulus P, Rigo P, Fillet G. Whole-body positron emission tomography using 18F-fluorodeoxyglucose compared to standard procedures for staging patients with Hodgkin's disease. Haematologica. 2001 Mar;86(3):266-73. PMID: 11255273.
  5. Hasenclever D, Diehl V. A prognostic score for advanced Hodgkin's disease. International Prognostic Factors Project on Advanced Hodgkin's Disease. N Engl J Med. 1998 Nov 19;339(21):1506-14. doi: 10.1056/NEJM199811193392104. PMID: 9819449.
  6. International Non-Hodgkin's Lymphoma Prognostic Factors Project. A predictive model for aggressive non-Hodgkin's lymphoma. N Engl J Med. 1993 Sep 30;329(14):987-94. doi: 10.1056/NEJM199309303291402. PMID: 8141877.
  7. Greenberg PL, Tuechler H, Schanz J, et al. Revised international prognostic scoring system for myelodysplastic syndromes. Blood. 2012;120(12):2454-2465. doi:10.1182/blood-2012-03-420489
  8. Hasford J, Baccarani M, Hoffmann V, Guilhot J, Saussele S, Rosti G, Guilhot F, Porkka K, Ossenkoppele G, Lindoerfer D, Simonsson B, Pfirrmann M, Hehlmann R. Predicting complete cytogenetic response and subsequent progression-free survival in 2060 patients with CML on imatinib treatment: the EUTOS score. Blood. 2011 Jul 21;118(3):686-92. doi: 10.1182/blood-2010-12-319038. Epub 2011 May 2. PMID: 21536864.
  9. International CLL-IPI working group. An international prognostic index for patients with chronic lymphocytic leukaemia (CLL-IPI): a meta-analysis of individual patient data. Lancet Oncol. 2016 Jun;17(6):779-790. doi: 10.1016/S1470-2045(16)30029-8. Epub 2016 May 13. PMID: 27185642.

Point 4: Methods: In the statistical analysis, the authors state that they first used univariate followed by multivariate logistic regressions. Later, they state they used Cox regression and report HRs. The terminology appears confusing to me: 1) The authors should check whether they performed a multivariate or multivariable regression (see 10.2105/AJPH.2012.300897). 2) Logistic regression yields ORs and does not comprehend the full time information of a time to event analysis, whereas Cox regression incorporates this time information to report HRs. I suggest that the authors disentangle the terminology in their analysis to describe what type of analysis they performed.

Response 4: Please accept our apologies that terminology has been mixed-up. This paper does not contain any Cox regression models. Only logistic regression (LR) models have been used for risk factors identification. We have updated statistical methods paragraphs together with table 3 and table 4. The updated version of text is as follows:

“Kolmogorov–Smirnov test was used to assess the normality of the distribution. A Student’s t-test was used to evaluate the differences between the two independent normally distributed variables, while the Mann–Whitney U test was used for non-normally distributed variables. Fisher’s exact test was used to evaluate the differences between two independent categorical data groups. Univariate and multivariate logistic regression models were used to evaluate the potential risk factors for ICU mortality. Factors that were found to be statistically significant in the univariate model were subsequently entered into a multivariate logistic model using a stepwise factor selection process. Odds ratios and their 95% confidence intervals were also calculated. Youden’s J statistic with receiver operating characteristic (ROC) analysis was used for cut-off values. Survival was estimated using the Kaplan–Meier method. A two-tailed p-value less than 0.05 was considered to be statistically significant. Statistical analysis was performed using the Statistical Analysis System (SAS Institute, Cary, North Carolina, USA) package version 9.2.”

Point 5: Methods: The authors indicated that they used stepwise regression with backward selection. Why did the authors not simply do a multivariable regression (which might be sufficient)? In other words, what is the advantage of using the complex stepwise regression approach? By that end, when the authors stick to the stepwise regression, I suggest that the authors report which variables were omitted, what the goodness of fit of the model is, and how they addressed the issue of overfitting? On a side note, do two variables in Table 4 (equal SOFA and increased SOFA) have exactly the same p value?

Response 5: Thank you for opportunity to explain our approach to logistic regression. We used stepwise selection method in order to find factors with the strongest impact on ICU mortality. This process of variable selection excludes factors with weak impact and minimizes random weak factors impact in the model. In order to reduce the chances of overfitting we decided to split multivariate LR analysis into 2 steps. In the first step we have grouped factors which were found to be statistically significant in univariate LR into three groups: scoring systems (qSOFA score ≥ 2, SOFA score ≥ 10, APACHE II score, SAPS 3 score, increase in SOFA score over 48 hours in ICU), therapeutic (renal replacement therapy, continuous renal replacement therapy, mechanical ventilation on the first day in ICU, mechanical ventilation anytime in ICU, vasopressors anytime in ICU, noradrenaline first day in ICU, maximum dose of noradrenaline, colistin therapy in ICU) and laboratory (potassium concentration on arrival to ICU, hemoglobin concentration on arrival to ICU, apH on arrival to ICU, base excess on arrival to ICU, lactate on arrival to ICU). Multivariate LR has been performed in each group and only factors which remained to be statistically significant have been selected and entered in the final (step 2) multivariate LR which is presented in results.

Responding to question regarding p values in Table 4 for “equal SOFA” and “increased SOFA”. Yes, this is correct that p value is 0.0156 because it is the value of general model.

Point 6: Methods/Results: The flowchart in Figure 1 and characteristics in Table 1 should be moved to the results section.

Response 6: Thank you for this suggestion. Figure 1 and Table 1 have been moved to results section.

Point 7: Methods: Last-patient-in was reported to be Dec 2019, and mortality is reported as of 28 May 2020, which is less than 365 days after; however, the authors report a 365-day mortality. How many patients were followed for 365 days and how many patients less than that? I suggest to either report the number of patients followed <365 days or leave out the 365-days mortality (e.g., in favor of 6-months mortality if the authors have data on this from almost every patient).

Response 7: Thank you for your comment. There are 85 patients (74.56 %) who were followed for more than 365 days, while follow up of 29 patients (25.44%) patients is less than 365 days. We don’t have 6-months data for 5 patients, but we have 100 % data for 90 days. Also we focus on ICU mortality, factors associated with it.  We agree with you that we should report survival only of the patients all of which are followed for that period of time. We removed fields beyond the 90-day mark from Figure 1. Study flow diagram. We also changed information from 365-day mortality to-90 day mortality in results section and discussion in order to maintain continuity.

Point 8: Discussion: At the beginning of the discussion, the authors compare their mortality rates with the rates from other studies. Did the other studies also consider only onco-hematological ICU patients? Were those multi-center, prospective studies? How many patients were analyzed? To put the numbers into context, the reader needs more information about the studies that the authors’ results are compared to.

Response 8:  Thank you for pointing this out. All of these studies included only patients with hematological malignancies. We included information about the studies which are discussed and updated manuscript text as follows:

“These results are similar to previous studies and mortality is lower than it was in multicenter retrospective study of 200 patients with hematological malignancies made in Poland (ICU mortality, 67%; hospital mortality, 78.8%) [10] and single centre prospective observational study of 170 patients with hematological malignancies treated in ICU in Croatia (ICU mortality, 53.5%) [11], but the reported data are limited to only a few centres. Better results were found in a prospective study of  1,011 critically ill patients with hematologic malignancies from 17 centres in France and Belgium; hospital mortality was 39.3%, and 90-day mortality - 47.5% [18].”

Point 9: Disucssion: The authors state that other studies examined the effect of higher noradrenaline doses on mortality (references 23-25). Were these studies on onco-hematological patients only, or on general ICU patients?

Response 9: We are very sorry because we made a confusing  mistake when transfering text with Zotero into journals template and studies indicated with references 23-25 are not in the right place in manuscript. We updated the text and removed references which appeared not in the right place. The statment „There are studies which analysed the effect of higher noradrenaline dose“ is true and we meant studies which are presented in further text. The information regarding patient status in terms of hematological malignancies was already  provided in the text and we did not changed it. Ir remained as follows:

“There are studies which analysed the effect of higher noradrenaline dose. For patients with septic shock and noradrenaline dose ³ 0.3 μg/kg/min 28-day mortality varied from 48.5% to 72.2%, only 17% of patients had hematological comorbidity [30].  In another study patients received a vasopressor dose of at least 1 μg/kg/min for more than 1 h, and a cut-off value associated with mortality was 0.75 μg/kg/min. Additionally, 35% of patients were immunocompromised, but the authors do not provide data regarding hematological malignancies [31]. It was found that all critically ill patients with febrile neutropenia who received a noradrenaline dose that was higher than 0.1 μg/kg/min died in the ICU [3].”

Minor issues:

Point 10: Figure 1: In the upper two boxes, the authors state “patients admitted to hospital/ICU”, but as re-admissions are included, we suggest to re-name this as “cases admitted”.

Response 10: Thank you very much. We are very thankful for this accurate suggestion. We have made changes and updated Figure 1.

Point 11: Figure 1: We suggest to re-arrange the mortality boxes and include them in the flowchart (the way they are arranged right now leaves them unrelated to the patient flow): There should be a vertical arrow from the box “original cases” down to a new box “survived at least 365 days” (or similar). The mortality boxes should then indicate how many patients leave the flowchart along this way (with horizontal arrows to each mortality box).

Response 11: Thank you for this meaningful suggestion. We removed outcome boxes which were beyond 90-day mark. We added arrows. We also changed mortality to survival and added the boxes which indicate the patients who died during that period.

Point 12: Table 1: The authors report mean APACHE II, SAPS III and qSOFA scores, but do they represent the mean over all measurements (multiple measurements per patient)? This requires explanation in the Table or its caption.

Response 12: Thank you for addressing this point. SAPS 3, APACHE II and qSOFA scores were calculated only once. We updated Table 1 caption. With the following text:

“SAPS 3 and APACHE II scores were calculated within the first 24 hours after admission to the ICU. If the FiO2 was ≥50%, we calculated alveolar–arterial gradient using geographical atmospheric pressure for Vilnius, which was 750.025 mmHg (114 m altitude above sea level and temperature of 22°C, which was room temperature in the ICU). QSOFA was calculated on admission to ICU.”

Point 13: Results: I assume that the authors selected the optimal cut-off noradrenaline dose based on the highest Youden index. If so, the authors should state this in the results section (and also report the value of the Youden index).

Response 13: Thank you for addressing this point. We made it more accurate and updated results section which now states that: “The optimal cut-off value (Youden Index = 91) for the noradrenaline dose was 0.212 μg/kg/min”.

In addition to the above comments, we updated the list of references.

Best regards,

Sarunas Judickas

Reviewer 2 Report

Thank you for reading the article. Very interesting article. I propose to divide the methodology into sub-sections: design, participants, research organization, statistical analysis, etc. There were also no research constraints and therapeutic implications. 

Author Response

Response to Reviewer 2 Comments

Dear Sir or Madam,

We appreciate the time and effort that you have dedicated to providing your valuable feedback on our manuscript. We are grateful to you for your insightful comments on our paper. We have been able to incorporate changes to reflect the suggestions provided by you. The changes made within the manuscript are seen in “Track changes”. Here is a point-by-point comments and concerns.

Point 1: Thank you for reading the article. Very interesting article. I propose to divide the methodology into sub-sections: design, participants, research organization, statistical analysis, etc. There were also no research constraints and therapeutic implications.

Response 1: Thank you for great suggestion. We updated methodology text with subsections. We also added the therapeutic implication to the conclusions, which is as follows:

“The ICU mortality of oncohematological patients in our study was 44.74%. Factors that were independently associated with it were qSOFA score ≥2, increase in the SOFA score over the first 48 hours in ICU, need for invasive mechanical ventilation on the first day in the ICU, need for colistin therapy and lower arterial pH upon arrival in the ICU. We found that a low cut-off value for the noradrenaline dose was associated with ICU mortality. Thus, we suggest that condition of oncohaematological patients is more critical than it appears and these patients need to be transferred to ICU before it becomes too critical.”

Best regards,

Sarunas Judickas

Round 2

Reviewer 1 Report

Thank you for giving me the opportunity to review the revised version of the manuscript. The authors were very responsive to the comments made and their changes significantly improved the manuscript’s quality. Three small issues remain, which I have detailed below.

Remaining issues:

  1. Methods, Results: I would like to thank the authors for revising the statistics section, which substantially gained clarity. I have one remaining comment on terminology, though: The regression that the authors present in Table 4 is, from my perspective, a multivariable instead of a multivariate regression (see doi:2105/AJPH.2012.300897). I would like to invite the authors to briefly respond to this comment or change the terminology.
  2. Methods: Thank you for deliberating on the details of your statistical model, which helps me understand your approach. That being said, I believe that the manuscript would benefit from a short description (1-2 sentences) on your 2-step approach to build the multivariable regression model (what you have detailed in your rebuttal).
  3. Results: Just a minor comment, but please make sure you refer to Fig. 1 at some point in the text.

Author Response

Response to Reviewer 1 Comments

Dear Sir or Madam,

We appreciate the time and effort that you have repeatedly dedicated to providing your valuable feedback on our manuscript. We are very happy to see our manuscript improved with your help and we want to thank for lessons we’ve learned. We are grateful to you for your remaining insightful comments on our paper. We have been able to incorporate changes to reflect the suggestions provided by you. The changes made within the manuscript are seen in “Track changes”. Here is a point-by-point response to comments.

Remaining issues:

Point 1: Methods, Results: I would like to thank the authors for revising the statistics section, which substantially gained clarity. I have one remaining comment on terminology, though: The regression that the authors present in Table 4 is, from my perspective, a multivariable instead of a multivariate regression (see doi:2105/AJPH.2012.300897). I would like to invite the authors to briefly respond to this comment or change the terminology.

Response 1: Thank you for your help and comment. Thank you for reference regarding regression. Yes, you are absolutely right. We have incorporated your suggestion throughout the manuscript and changed the terminology.

Point 2: Methods: Thank you for deliberating on the details of your statistical model, which helps me understand your approach. That being said, I believe that the manuscript would benefit from a short description (1-2 sentences) on your 2-step approach to build the multivariable regression model (what you have detailed in your rebuttal).

Response 2: Thank you for your insight and valuable details regarding statistics. We are very happy to update text with 2-step approach that we used. We updated manuscript with the following text: “Multivariable logistic regression was performed with two steps: first – we grouped factors which were found to be statistically significant in univariate logistic regression into three groups: scoring systems, therapy and laboratory tests. After that multivariable logistic regression has been performed in each group and only factors which remained statistically significant have been selected and entered in the final (second step) multivariable logistic regression which is presented in results”.

Point 3: Results: Just a minor comment, but please make sure you refer to Fig. 1 at some point in the text.

Response 3: Thank you for pointing this out. The Figure 1 is mentioned in section 2.3. Research data (line Nr. 93).

Best regards,

Sarunas Judickas

Reviewer 2 Report

thank you for the amendments made, I have no comments 

Author Response

Response to Reviewer 2 Comment

Dear Sir or Madam,

We appreciate the time and effort that you have repeatedly dedicated to providing your valuable feedback on our manuscript. We are very happy to see our manuscript improved with your help and we want to thank for lessons we’ve learned.

Best regards,

Sarunas Judickas